# The Influence of Mixed Reality on Satisfaction and Brand Loyalty in Cultural Heritage Attractions: A  Brand Equity Perspective

**Sujin Bae [1], Timothy Hyungsoo Jung [2], Natasha Moorhouse [2], Minjeong Suh [3] and Ohbyung Kwon [4,\*]**

[1]  School of Management, Kyung Hee University, Seoul 02447, Korea; luckybsj@khu.ac.kr
[2]  Faculty of Business and Law, Manchester Metropolitan University, Manchester M15 6BG, UK; t.jung@mmu.ac.uk (T.H.J.); n.moorhouse@mmu.ac.uk (N.M.)
[3]  Graduate school of international tourism, Hanyang University, 222, Wangsimni-ro, Seongdong-gu, Seoul 04763, Korea; lucymmu@naver.com
[4]  School of Management, Kyung Hee University, Kyung Hee Dae Ro 26, Dong Dae Mun Gu, Seoul 02447, Korea
\*  Correspondence: obkwon@khu.ac.kr; Tel.: +82-2961-2148

**Abstract:** Mixed reality technology is being increasingly used in cultural heritage attractions to enhance visitors' experiences. However, how the characteristics of mixed reality affect satisfaction and brand loyalty has not been explored in previous research.  The purpose of this study is to identify factors affecting satisfaction with mixed reality experiences at cultural and artistic visitor attractions and their influence on brand loyalty, which is connected with management performance. We propose a  theoretical model based on brand equity theory in the context of mixed reality experience. Survey data were gathered from 251 respondents visiting a cultural and artistic visitor attraction in Seoul, Korea, using a stratified sampling method.  Partial least squares structural equation modeling (PLS-SEM) was employed for the data analysis.  The results suggest that the characteristics of mixed reality (interactivity, vividness) not only influence the affective aspects (perceived immersion, perceived enjoyment) of visitors' experiences, but also positively affect brand awareness, brand  association, and brand loyalty.

**Keywords:** mixed reality; interactivity; vividness; brand loyalty; brand awareness; brand association; cultural heritage attractions

## 1. Introduction

As 5G provides very high data rates, extremely low latency, and significant improvement in users' perceived quality of service (QoS), it is expected to become more widespread in the coming years particularly in urban areas [1]; hence, interest in developing mixed reality content with high user acceptance is increasing. Mixed reality content has traditionally been used for games [2], education [3], training [4], tourism [5], and cultural exhibitions [6]. Satisfaction with cultural and artistic exhibitions has been linked to the consumption of souvenirs and visitor attractions. Accordingly, researchers have examined the factors affecting construction and operation of exhibitions by public institutions and private companies [7]. Case studies focused on planning, composition, and exhibition content have demonstrated that visitor satisfaction with exhibition experiences can be increased by adding various information technologies, including augmented reality (AR), virtual reality (VR), and holograms. Integration of mixed reality technology into cultural and artistic attractions is expected to contribute to a richer visitor experience [8].

As with any other organizations, the sustainability of cultural and artistic visitor attractions consists of four elements including the cultural environment (e.g., heritage preservation, new audiences), social environment (e.g., engagement, social responsibility), natural environment (e.g., green technologies, energy efficiency), and economic environment [9]. Economic aspects can be supported by increasing visitor numbers and net growth [10]; hence, when considering factors that facilitate the continued operation of cultural and artistic visitor attractions, it is necessary to study factors affecting visitor satisfaction [11] and souvenir purchase [12]. This information is of interest to corporate executives and practitioners. However, empirical research is lacking on the characteristics of mixed reality and their connection to user satisfaction and brand loyalty in the context of cultural and artistic visitor attractions. Although one study examined development of the experimental environment and proof of concept [13], the effects of the technical characteristics of mixed reality on satisfaction or purchase intention have rarely been studied [14].

The marketing efforts of cultural organizations tend to be focused on sustaining and creating loyalty [15], which can be affected by brand equity [16–18]. Customer-based brand equity (CBBE), which can be defined as the added value of a brand to a customer [19], is said to reinforce brand loyalty and brand preference, attract new customers, and generate greater resiliency to competitors' marketing actions, enabling the brand to earn consistent, predictable, and higher market shares over time [20]. According to early research [21,22], brand awareness, brand association, and brand loyalty are three key attributes of brand equity that must be perceived positively by the consumer for brand equity to be achieved. Hence, brand equity is formed when a brand known to the consumer is perceived as positive and unique with strong associations [23]. For organizations, brand equity is important because of its ability to influence positive affect and attitudes, which manifest themselves in consumer behaviors such as purchase intention and repeat visitation [23]. Brand equity theory has been utilized in research conducted in both cultural (e.g., [24,25]) and technological contexts (e.g., [19]). However, to date, there is limited research employing a brand equity perspective to investigate the influence of visitors' mixed reality experience on brand equity in the context of cultural and artistic visitor attractions through empirical research using quantitative approach. Therefore, this study aims to bridge this research gap, and as a result, it offers important implications for managers of such organizations in addition to making a theoretical contribution to research in this area.

More specifically, the purpose of this study is to examine how two characteristics of mixed reality (interactivity, vividness) affect the affective aspects (perceived immersion, perceived enjoyment) of visitors' experiences of and satisfaction with cultural and artistic attractions. In addition, we investigate the effects of the characteristics of mixed reality on brand awareness, brand association, and, eventually, brand loyalty, which is connected to the management of cultural and artistic visitor attractions. We use brand equity theory as the theoretical background and conduct an empirical analysis with visitors to a specific cultural and artistic attraction in which mixed reality has been implemented.

This paper begins with a theoretical background exploring previous literature on the implementation of mixed reality in cultural heritage attractions, followed by an overview of brand equity theory. A justification for each of the proposed hypotheses is then presented, which is followed by the method detailing the quantitative data collection and analysis. Finally, discussions and conclusions including theoretical contributions and practical implications are offered.

## 2. Theoretical Background

### 2.1. Mixed Reality in Cultural Heritage Attractions

Cultural heritage-related curated spaces, including art galleries and museums, compete for cultural visitors' time. They are under pressure to create appealing and individual experiences that attract visitor's attention [26]. Immersive technologies such as mixed reality (MR), AR, and VR have become increasingly used in cultural heritage attractions, providing experiences through a combination of real and digital content [27]. Several studies focus on the application and use of both AR and VR in

urban cultural heritage attractions. These studies are written from various perspectives, including the impacts of AR on urban heritage tourism (e.g., [28]), value of AR from stakeholders' perspectives (e.g. [29]), experiential learning with AR in museums (e.g., [30,31]), AR acceptance in urban heritage tourism (e.g., [32,33]), and experiencing AR and VR in art galleries and museums (e.g., [34,35]). By comparison, fewer studies have focused on mixed reality (e.g., [5,36]), which can be defined as "the combination of virtual computer graphics and real-world elements into a virtual environment" ([26] p. 89). Powered by mobile devices or HMD(Head Mounted Display)s, mixed reality can enrich the user's experience by providing enhanced information through multisensory modalities such as video, audio, touch, and scent. Mixed reality enhances the user's experience of the surrounding environment.

Two characteristics of mixed reality can enhance users' experience of a cultural heritage attraction. Feelings of presence tend to occur when a virtual environment provides high levels of interactivity and vividness [37–39]. Interactivity allows users to manipulate the virtual environment in real time [40] and is perceived to be one of key determinants of enjoyment in interactive systems [41]. Vividness (i.e., "the ability of a technology to produce a sensorially rich mediated environment"; [39] p. 80) and interactivity are two important media features of mixed reality. In AR research, vividness and interactivity have been found to influence consumer evaluations positively through increased immersion [42]. In addition, AR offers increased interaction through 3D virtual objects, and system quality has been found to enhance learning effectiveness and learner satisfaction with the learning environment [43].

By connecting information and material reality, mixed reality can make the invisible visible in situ, allowing visitors to experience the space, its history, and a community simultaneously [26]. The stimulus-organism-response (S-O-R) model [44], which asserts that a stimulus (S) may influence an organism (O) that in turn influences behavior (R), was one of the first models developed to investigate the influence of atmosphere on customer behaviors; this approach continues to be used widely today [45] in immersive technology research (e.g., [46]), among other fields of study. As mixed reality technologies continue to advance, their potential applications in hybrid spaces increase and become more widespread [26]. Although researchers have begun to investigate how mixed reality can be integrated into urban cultural heritage experiences (e.g., [14]), research in this area is largely still in its infancy, and no comprehensive framework has been developed for evaluating these novel experiences and their multiple dimensions of adoption, usability, engagement, immersion, presence, and physical awareness [26]. As highlighted by Trunfio and Campana [47], further investigation is also required into the influence of mixed reality on visitor experience and value propositions in such spaces.

## 2.2. Brand Equity Theory

Brand equity has been an important marketing concept since the 1980s [48] and has recently become an increasingly important concept for cultural organizations [49]. Many studies on customer-based brand equity (CBBE) rely on the models created by Aaker [21] and Keller [22,50]. Although these two authors conceptualized brand equity differently, they both generally defined the concept from a consumer-based perspective [51]. Brand equity is generally considered a multidimensional construct [19]. According to Aaker, it comprises brand awareness, brand association, perceived quality, brand loyalty, and other proprietary assets (e.g., patents, registered trademarks, and so on), which must be perceived positively by the consumer for brand equity to be achieved [21,52]. Brand awareness plays an important role in CBBE [22]; it refers to a) consumers' ability to recall, recognize and connect with a brand (via brand name, logo, symbol, etc.), linking it with certain associations in memory (depth of brand awareness), and b) the range of purchase and consumption situations in which the brand comes to mind (breadth of brand awareness) [53]. To create brand equity, it is also important that the brand has strong, favorable, and unique brand associations [22]. Brand association is another component of brand equity linked to consumers' memory of a brand [54], which provides the basis of purchase decision-making [55]. Favorable and unique brand associations are essential to brand image [56]; therefore, when examining brand association, many studies focus on brand

image (e.g., [49,57]). Overall, positive brand association and significant brand awareness contribute to a positive consumer response [19].

Empirical studies have demonstrated that the higher the brand equity, the greater the loyalty [16–18]. A brand is likely to have positive brand equity if it receives more favorable responses, that is, higher levels of brand association, awareness, preference, and familiarity [55]. The role of the brand in creating preferences and generating value and loyalty is no doubt of interest to managers of cultural institutions [49]. Indeed, most of their marketing efforts are directed toward sustaining and increasing loyalty [15]. Research shows that when a museum's brand is evaluated positively, customer loyalty toward the museum is greater [25]. Although CBBE has been widely discussed in tourism research [25,52,58], there has been little application of this approach to urban cultural heritage organizations, with only a few researchers investigating CBBE in the museum [24,25] and art contexts [49] and even fewer focusing on mixed reality in urban cultural heritage attractions from this perspective.

## 3. Hypotheses

In this section, we present our hypotheses. Figure 1 shows our theoretical model, which represents the characteristics of mixed reality affecting visitor satisfaction via perceived immersion and perceived enjoyment and the associated hypotheses. Based on brand equity theory, we also posit that the characteristics of mixed reality affect brand awareness, and in turn, brand awareness influences brand association and brand loyalty.

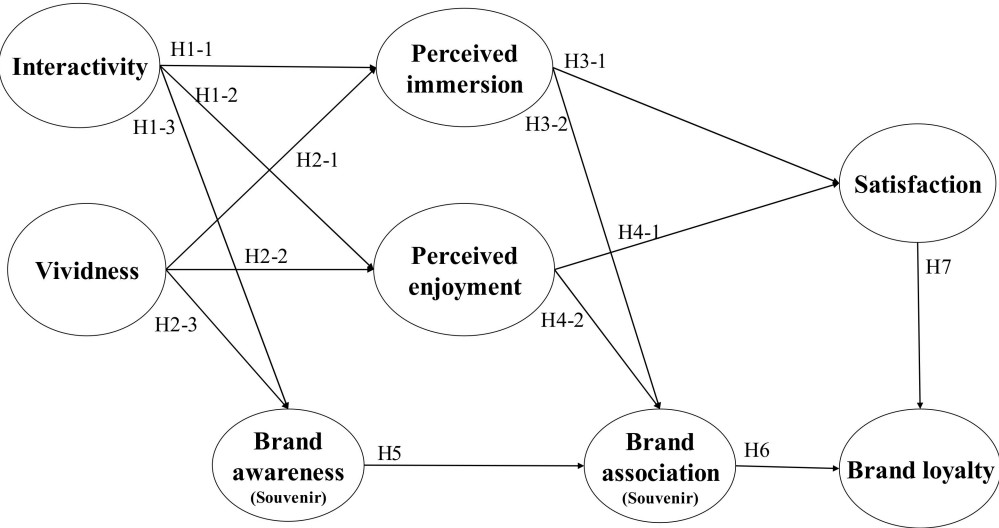

**Figure 1.** Research model.

*3.1. Interactivity, Perceived Immersion, Perceived Enjoyment, and Brand Awareness*

Interactivity refers to the degree to which users of a medium can influence the form or content of the mediated environment [59]. Some research suggests that interactivity can have an impact on the personal experience of the user [59]. User experience consists of perceived immersion and perceived enjoyment. Interactive games are perceived to be immersive when they offer predictable, tightly scripted interactions that enable the user to enjoy the virtual experience [60]. Early research indicates that interactive components, multisensory stimulation, and dynamic displays influence immersion and are salient and important design features that distinguish one area from another in the mind of the visitor [61]. Later, Shafer et al. [62] found that perceived reality influenced predicted spatial presence, which in turn was a significant predictor of enjoyment. Overall, a more immersive, realistic, and fun experience can be achieved through a more natural user interface [62].



Previous research also suggests a positive relationship between interactivity and perceived enjoyment [62–64]. For instance, in Lin and Parker's [63] study, interactivity was considered the most profound way to enhance the user experience in terms of increased presence and enjoyment in virtual environments. In the context of motion-based systems (e.g., Kinect), Shafer et al. [62] found similar results: higher interactivity increased spatial presence, perceived reality, and enjoyment. Interactivity in artistic spaces has also been found to increase audience enjoyment and creative identity [64]. Finally, Shin [65] proposed a model of attitudes towards 3DTV, identifying significant roles for social presence and flow, both of which affect attitude, perceived usefulness, and perceived enjoyment. Accordingly, these factors are considered to underlie users' expectations of 3DTV [65].

When evaluated as characteristics of an online brand community, both virtual interactivity and information quality were found to have strong effects on customer engagement [66]. Customer engagement is a strong predictor of brand loyalty. As a result, many organizations are developing brand communities to promote and communicate their offerings to customers [66]. Recent research demonstrated that the more actively firms use and manage social media, the higher awareness of the brand will be [67]. Moreover, Mauroner et al. [68] found that interactivity has a significant influence on brand recall, suggesting that interaction with an ad may lead to more intense cognitive processing of the brand message. Hence, we hypothesize the following:

**H1-1:** *Interactivity of mixed reality will have a positive effect on perceived immersion.*

**H1-2:** *Interactivity of mixed reality will have a positive effect on perceived enjoyment.*

**H1-3:** *Interactivity of mixed reality will have a positive effect on brand awareness.*

*3.2. Vividness, Perceived Immersion, Perceived Enjoyment, and Brand Awareness*

Vividness, also known as realness, realism, or richness [69,70], refers to the ability of a technology to produce a sensorially rich, mediated environment ([39] p. 80). It includes not only the sensory experience of actual objects, but also the non-sensory experience of imaginary objects [71]. Vividness is also perceived as a quality of brand expression rather than a quality of the brand itself. For example, in e-commerce, it is known that how a product is displayed on the web affects evaluation and sales of that product [72].

Vividness has a positive effect on immersion in the VR context [42]. To feel immersed, consumers must be free to interact with products, to inspect vividly and realistically generated virtual product images [73]. This relationship between vividness and immersion is likely also to be present in mixed reality, where virtual objects appear as a vivid and seamless part of the actual physical scene.

On the other hand, vividness may also have a positive effect on the enjoyment of mixed reality [74]. In VR, vividness is recognized as a representative form of modality richness, which is known to have a significant effect on shopping enjoyment [75]. When it comes to e-commerce, the visual vividness of online websites is associated with enjoyment [76]. Similarly, McLean and Wilson [77] suggested that perceived enjoyment obtained from technology is related to two functional elements: interactivity and vividness. We believe this will also be true in the mixed reality context.

According to earlier research, extrinsic cues of website vividness allow unfamiliar brands to compete against familiar brands [78]. Highly vivid advertisements have been found to influence positive brand attitudes compared with less vivid advertisements, which in turn stimulates consumers' purchase intentions [75]. Additionally, a positive correlation has been demonstrated between the vividness of word of mouth communication messages and both brand awareness and brand association [62]. Therefore, we present the next hypothesis as follows:

**H2-1:** *Vividness of mixed reality will have a positive effect on perceived immersion.*

**H2-2:** *Vividness of mixed reality will have a positive effect on perceived enjoyment.*

**H2-3:** *Vividness of mixed reality will have a positive effect on brand awareness.*

*3.3. Perceived Immersion, User Satisfaction, and Brand Association*

Perceived immersion refers to the degree to which a person is immersed in the content being experienced [61,79]. Immersion is an important factor in game experience. Denisova and Cairns [80] found that the difference between an adaptive AI group and a standard AI group at the level of the user reflects the specific state in which game users experience immersion. Immersion, passion, and activation are key elements of consumer brand engagement that represent the degree to which a customer is prepared to exert relative cognitive, emotional, and behavioral resources in specific interactions with a focal brand [81]. Highly immersive environments that reproduce perceptual richness, thus heightening the sense of presence, have positive effects on self-endorsement [82]. Self-endorsement subsequently triggers favorable attitudes toward products and brands. Prior research suggests that users wearing a particular brand of clothing in a highly immersive environment prefer the brand worn by their virtual self to the brand worn by others [83].

High sensory immersion is also associated with a more favorable brand attitude, greater product knowledge, and elevated purchase intention [83]. In the context of 3D virtual worlds, presence was found to influence enjoyment and brand equity, which is consistent with the theory of flow, which states that a sense of immersion can produce hedonic outcomes and influence learning and attitudes [19]. Even when AR technology is applied to a game, perceived immersion affects the satisfaction of game participants [84]. In addition, perceived immersion has a positive effect on satisfaction in a merchant virtual universe, which hyper-realistically reproduces the environment of a real shopping mall [85]. Likewise, perceived immersion affects satisfaction with experience. We surmise that the same results may be possible in the context of mixed reality experiences. Therefore, we state the next hypothesis as follows:

**H3-1:** *Perceived immersion in mixed reality experiences will have a positive effect on user satisfaction.*

**H3-2:** *Perceived immersion in mixed reality will have a positive effect on brand association.*

*3.4. Perceived Enjoyment, User Satisfaction, and Brand Association*

In the user experience domain, researchers either focus on one or two factors (e.g., previous experience or interactivity, involvement and vividness, intention or presence, enjoyment and simulator sickness) (e.g., [66,75,86]). For instance, Islam and Rahman [66] investigated how the unique characteristics of a brand community, including information quality, system quality, virtual interactivity, and rewards, affect customer engagement and, subsequently, brand loyalty. Van Kerrebroeck et al. [75] explored the impact of VR on transformational brand appeals and found an association between this technology and higher perceptions of vividness and presence than regular 2D video. Finally, Virvilaite et al. [86] studied the relationship between vividness and usefulness (word of mouth communication messaging) and four brand equity components (brand association, brand awareness, brand loyalty, and perceived quality) and found an overall positive effect.

Perceived enjoyment refers to the degree to which visitors who experience mixed reality content enjoy it [87,88]. According to the expectation-confirmation model of Shiau and Luo [89], perceived enjoyment has a positive effect on satisfaction. In addition, another study using the expectation-confirmation model by Joo et al. [90] confirmed that perceived enjoyment has a significant positive effect on satisfaction and sustained use intention for students using digital textbooks. In another study on learning outcomes of VR users in education, it was found that the participants' perceived enjoyment of the VR experience not only affected perceived learning outcomes, but also that perceived learning outcomes positively affected satisfaction, perceived learning, and behavioral intention [91]. Prior research has indicated that enjoyment of an advertisement can affect people's attitude towards the advertised product [92]. Hence, we conclude that perceived enjoyment is related to attitudes towards new marketing systems. Recent research supports this assertion that perceived enjoyment positively

influences brand attitude [93]. Likewise, in the context of mixed reality experience, perceived enjoyment may affect satisfaction. Therefore, we present the next hypothesis as follows:

**H4-1:** *Perceived enjoyment of mixed reality content will have a positive impact on user satisfaction.*

**H4-2:** *Perceived enjoyment of mixed reality content will have a positive impact on brand association.*

### 3.5. Brand Awareness and Brand Association

Brand awareness is a key determinant of brand equity [20,94–96]. It is defined as an individual's ability to recall and recognize a brand [97]. In the context of this study, brand awareness refers to whether consumers can recall or recognize a brand or simply whether they know about a brand at all [95]. Brand awareness indicates an association in memory about a particular brand [98]; brand awareness can have either a positive or negative impact on brand association [99]. This is observed in many brands, including car brands [100]. Therefore, we hypothesize as follows:

**H5:** *Brand awareness of souvenirs in galleries with mixed reality content will have a positive impact on brand association.*

### 3.6. Brand Association and Brand Loyalty

As previously mentioned, we investigate brand awareness, brand association, and brand loyalty from the perspective of brand equity theory. The role of brand associations as an important element in brand equity management has been previously studied [101]. Brand association is closely related to customers' memories about a brand [96]. In brand equity theory, brand association is closely related to the degree of loyalty to a brand felt by customers [102]. This has been observed in several contexts, including an e-brand study [96]. In our study, brand association refers to everything related to the brand in consumers' minds. Brand association can help consumers determine the relevance of both the parent brand and extended brands [103]. We believe that brand association will affect brand loyalty in the context of cultural and artistic visitor attractions and mixed reality souvenir shops. Therefore, we propose the next hypothesis as follows:

**H6:** *Brand association for a souvenir in a gallery with mixed reality content will have a positive impact on brand loyalty.*

### 3.7. Satisfaction and Brand Loyalty

User satisfaction refers to visitors' degree of satisfaction with the content they experience [104]. In this study, brand loyalty refers to the customer's tendency to repurchase a brand, revealed through purchase behavior or brand sales (e.g., actual purchases observed over time) [105,106]. Research on user satisfaction and brand loyalty has been conducted in various contexts. For example, it was found that shopping mall visitor satisfaction has a positive effect on brand loyalty [107], and brand studies in the hotel and restaurant industries have found that brand experiences are driven by customer experience and satisfaction and have positive effects on loyalty [108]. However, there are few empirical studies that examine the effects on user satisfaction and brand loyalty of content display using mixed reality in the context of cultural and artistic visitor attractions. Similar to shopping and restaurant experiences, we expect that satisfaction with mixed reality in this context will have a positive impact on brand loyalty. Therefore, we hypothesize the following:

**H7:** *User satisfaction with mixed reality will have a positive effect on brand loyalty.*

## 4. Method

### 4.1. Study Context

For this study, we selected a cultural and artistic visitor attraction located in Seoul in an area where modern and classical culture and fashion encounter each other and are well balanced. The attraction, L'atelier, is a representative IT-based theme park which substantially and most successfully adopts interactive media including mixed reality technologies. Hence, L'atelier can be a good benchmark to investigate the viability of mixed reality technology for the sustainable operation of the cultural heritage attraction. L'atelier includes artwork and stories of various 19th-century Impressionist artists such as van Gogh and Gauguin [109]. It has recently integrated mixed reality technology to provide an immersive experience using multi-sensory features that contribute to a more interactive and enjoyable visitor experience. By using this technology, visitors can experience not only paintings but also hidden stories about various artists and the historical and cultural background relating to the paintings. Through AI integration, visitors are able to interact via conversation with friends of the impressionists such as van Gogh's friend, postman Joseph Roulin. They can also experience artwork through a media art show on the theme of "Monet's Garden" (e.g. visit 360 degrees, and models such as trees in the actual garden appear), which was facilitated by a physical display wall used to portray holograms and projection mapping technology. Monet depicted the water lily that changes with light; hence, the artist's garden in the media art show is the painting created by light. There is also a hologram talk show based on the mystery around the death of van Gogh and a musical performance, "van Gogh's Dream", which is based on his life story [109]. In the musicals, actors and characters in digital works interact with real actors through dialogue.

### 4.2. Measures

A survey with a total of 25 measurement items was used for this study. The following items were adopted from the literature: interactivity (e.g., "When I experienced Monet's garden, the mixed reality water lily leaves interacted with my movement.") [110], vividness (e.g., "When I experienced a musical, the digital characters felt vivid.") [111,112], perceived immersion (e.g., "While I was experiencing the mixed reality content such as a musical and Monet's garden, I was immersed in the mixed reality experience.") [84,113,114], perceived enjoyment (e.g., "I enjoyed my experience of mixed reality such as the musicals and Monet's garden.") [115,116], brand awareness (e.g., "When I went to L'atelier souvenir shop, I noticed a souvenir that reflects impressionist artwork.") [117], brand association (e.g., "When I looked at impressionist souvenirs, I could easily recall my mixed reality experience at L'atelier.") [118,119], satisfaction (e.g., "I am satisfied with the mixed reality content I experienced at L'atelier.") [120,121], and brand loyalty (e.g., "I am willing to purchase souvenirs from L'atelier souvenir shop.") [120,122]. The other items were identified through factor and reliability analyses and scored on a 7-point Likert scale. In order to avoid any material discrepancies in the survey, the initial questionnaire was created in English and then translated to Korean by researchers who are proficient in both languages.

### 4.3. Data Collection

For data collection, survey responses were gathered using a tablet PC. The researchers were situated in the café within the attraction on weekends between 21 and 29 September 2019, and survey data were gathered from visitors after they experienced mixed reality at the attraction. A total of six researchers were conducting survey data collection during this time. A stratified sampling method was used for data collection in order to match the demographic profiles (i.e., age and gender) of actual visitors to this cultural and artistic visitor attraction. More specifically, in order to identify the population and profile of visitors, researchers consulted with the marketing manager at the visitor and artistic cultural attraction in Korea. According to data gathered between November 2017 and October 2019, the majority of visitors to L'atelier are female (66% female, 33% male) aged between 20 and

30 years (86.7%), followed by those aged over 40 (13.3%). This was also proven during the pilot study. Based on this, sample data were collected by researchers using a stratified sampling method to match the population of actual visitors in terms of gender and age. Respondents experienced the attraction for one or more hours before visiting the souvenir shop. Each visitor was approached and invited to participate in the study at the end of their experience. Only those who volunteered to participate were included.

Data collection was conducted on weekends with an aim to gather an even number of respondents with regards to age and sex. The age range was limited to adults who were able to purchase their own entrance tickets. Visitors who had not fully experienced mixed reality during their visit were excluded from the survey. In order to recruit participants, one researcher wearing a uniform promoted the survey at the entrance while other researchers conducted the survey with visitors at the café near the exit. The researcher had a tablet PC dedicated to the questionnaire and the survey was conducted in the café. All participants were offered a gift box worth about USD 5 as compensation for participation. In total, 322 surveys were collected, of which 251 were used in the final analysis; others were excluded due to inappropriate answers to the reverse-coded questions. Each participant spent approximately 10 min completing the survey. SPSS 23.0 was used for demographic analysis and exploratory factor and reliability analyses. Table 1 shows the demographic characteristics of the participants, which were collected at the exit of the attraction using a stratified sampling method.

**Table 1.** Demographic characteristics of respondents (n = 251).

| | Category | Frequency (%) |
|---|---|---|
| **Gender** | Male | 84 (33.5) |
| | Female | 167 (66.5) |
| Age | 20s | 118 (47.0) |
| | 30s | 98 (39.0) |
| | 40s | 27 (10.8) |
| | 50s | 8 (3.2) |
| Education | High school graduate | 28 (11.2) |
| | College registration | 26 (10.4) |
| | College graduate | 168 (66.9) |
| | Graduate student or above | 29 (11.6) |
| Profession | Student | 49 (19.5) |
| | Employee | 126 (50.2) |
| | Housewife | 17 (6.8) |
| | Practitioner | 42 (16.7) |
| | Others | 17 (6.8) |

## 5. Results

Partial least squares structural equation modeling (PLS-SEM) was employed to test the hypotheses depicted in Figure 1. PLS-SEM was selected as it is designed to evaluate how well a proposed model or hypothetical construct explains the collected data. It is conducted using a two-step hybrid method, specifying a measurement model through confirmatory factor analysis, and then testing a latent structural model developed from the measurement model.

### 5.1. Factor Analysis

To create the questionnaire, 25 items were analyzed by exploratory factor analysis and the Varimax rotation method. The factor analysis (shown in Appendix A) revealed that commonality exceeded

0.683, and factor loadings revealed eight factors, with no multiple loading items for only one factor of 0.6 or more. The results of the exploratory factor analysis also revealed that the Kaiser–Meyer–Olkin (KMO) value for the sample was 0.905, which confirms that the dataset is valid for the purposes of factor analysis. In addition, the sphere formation test value for the sample was $x^2 = 8022.512$ (df = 300, $p < 0.001$), and the cumulative total variance of the factors was 90.39%, which also indicates suitability for factor analysis. Reliability of the eight identified factors was confirmed by Cronbach's $\alpha$ coefficient, which was higher than 0.832, thus indicating high credibility [123].

## 5.2. Validation

Validity and feasibility of the measurement model

The validity and appropriateness of the measurement model were determined before testing the hypotheses of this study. First, as shown in Table 2, the average variance extracted (AVE) exceeded 0.888, which indicates convergent validity [124]. Composite reliability (CR), which is an index that measures the feasibility of the measurement model, exceeds 0.900, indicating reliability. The Cronbach's $\alpha$ value resulting from the PLS algorithm was 0.833. Except for vividness, the internal consistency of questionnaire items was indicated by a high reliability value of 0.9 or higher on all other factors [124,125]. Furthermore, the communality value, measuring the quality of the measurement model, exceeded 0.888, indicating the suitability of the measurement model. The size of goodness-of-fit in the PLS path model is also regarded as large if the value is 0.36 or larger; the value for location-based AR was 0.536, thus showing high goodness-of-fit [126].

**Table 2.** Overall model fit.

| Variable | AVE | CR | $R^2$ | Cronbach's Alpha | Communality | Redundancy |
|---|---|---|---|---|---|---|
| Interactivity | 0.924 | 0.973 | | 0.959 | 0.924 | |
| Vividness | 0.749 | 0.900 | | 0.833 | 0.749 | |
| Perceived immersion | 0.888 | 0.959 | 0.288 | 0.936 | 0.888 | 0.158 |
| Perceived enjoyment | 0.925 | 0.974 | 0.279 | 0.959 | 0.925 | 0.113 |
| Brand awareness | 0.903 | 0.965 | 0.161 | 0.946 | 0.903 | 0.090 |
| Brand association | 0.950 | 0.983 | 0.332 | 0.974 | 0.950 | 0.136 |
| Satisfaction | 0.940 | 0.984 | 0.605 | 0.979 | 0.940 | 0.469 |
| Brand loyalty | 0.898 | 0.964 | 0.255 | 0.943 | 0.898 | 0.186 |
| (Goodness-of-fit) | MR = 0.536 *Quality: high (>0.36), medium (0.25~0.36), low (0.1~0.25) (Tenenhaus, Vinzi, Chatelin, and Lauro, 2005) | | | | | |

AVE: average variance extracted, CR: composite reliability; MR: model fit.

In order to ensure discriminant validity, the square root of AVE for each factor should be greater than the correlation coefficients between variables [127,128]. The results in Table 3 show that the model satisfies this condition. In addition, the cross-loading results in Appendix B show that all factor loading values exceeded 0.855, confirming the discriminant validity of each factor.

**Table 3.** Correlations among constructs.

| Constructs | Mean | SD | (1) | (2) | (3) | (4) | (5) | (6) | (7) | (8) |
|---|---|---|---|---|---|---|---|---|---|---|
| (1) Interactivity | 4.99 | 1.33 | **0.961** | | | | | | | |
| (2) Vividness | 5.43 | 1.08 | 0.356** | **0.866** | | | | | | |
| (3) Perceived immersion | 5.34 | 1.29 | 0.433** | 0.440** | **0.942** | | | | | |
| (4) Perceived enjoyment | 5.92 | 1.15 | 0.373** | 0.477** | 0.610** | **0.962** | | | | |
| (5) Brand awareness | 5.29 | 1.34 | 0.327** | 0.329** | 0.442** | 0.400** | **0.950** | | | |
| (6) Brand association | 5.09 | 1.44 | 0.334** | 0.400** | 0.527** | 0.465** | 0.388** | **0.975** | | |
| (7) Satisfaction | 6.12 | 0.99 | 0.369** | 0.554** | 0.650** | 0.734** | 0.438** | 0.564** | **0.969** | |
| (8) Brand loyalty | 4.86 | 1.57 | 0.395** | 0.301** | 0.452** | 0.422** | 0.616** | 0.414** | 0.470** | **0.948** |

Note 1: Values on the diagonal indicate the square root of AVE for each construct. Note 2: * $p < 0.05$, ** $p < 0.01$.

### 5.3. Hypothesis Testing

The hypotheses in this study were tested through path counting, and the valence of each path coefficient was confirmed by setting 5000 bootstrapping specimens [128,129]. The significance of individual paths is summarized in Figure 2. Thirteen out of thirteen paths exhibited a *p*-value less than 0.05. The explanatory power of the research model is also shown. The adjusted R-squared value shows that the constructs in the model together accounted for 60.5% of user satisfaction with their experience of mixed reality.

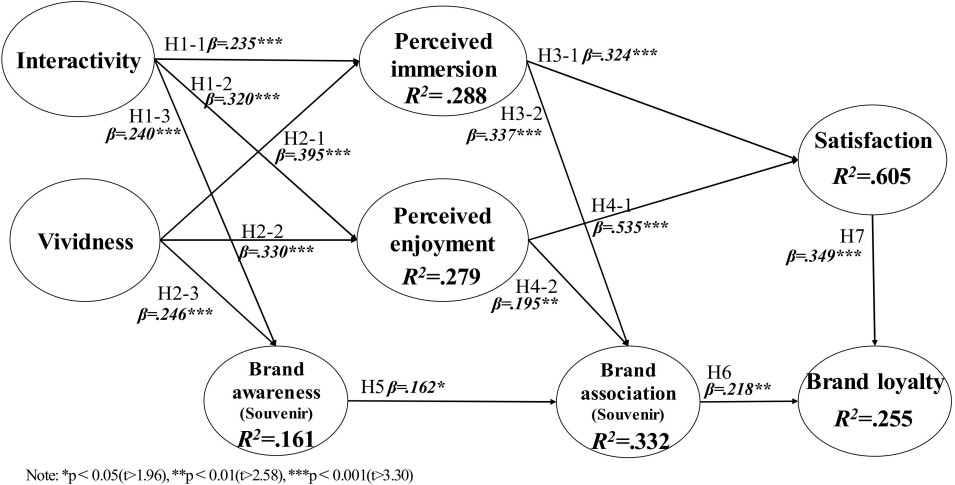

Note: *$p < 0.05(t > 1.96)$, **$p < 0.01(t > 2.58)$, ***$p < 0.001(t > 3.30)$

**Figure 2.** Hypothesis test results.

The results of hypothesis testing were as follows. First, values for interactivity during the mixed reality-based experience were significantly associated with perceived immersion (β = 0.320, t = 5.222; H1-1 was supported), perceived enjoyment (β = 0.235, t = 4.049; H1-2 was supported), and brand awareness (β = 0.240, t = 3.447; H6-3 was supported). Second, values for vividness during the mixed reality-based experience were significantly associated with perceived immersion (β = 0.330, t = 5.264; H2-1 was supported), perceived enjoyment (β = 0.395, t = 6.346; H2-2 was supported), and brand awareness (β = 0.246, t = 3.364; H2-3 was supported). Fourth, users' perceived enjoyment when engaging in the mixed reality-based experience was significantly associated with satisfaction (β = 0.535, t = 5.965; H4-1 was supported) and brand association (β = 0.195, t = 2.936; H4-2 was supported). Fifth, the users' perceived immersion when engaging in the mixed reality-based experience was significantly associated with satisfaction (β = 0.324, t = 3.614; H3-1 was supported) and brand association (β = 0.337, t = 4.394; H3-2 was supported). Sixth, brand association was significantly associated with brand loyalty (β = 0.218, t = 2.654; H6 was supported). Third, brand awareness was significantly associated with brand association (β = 0.162, t = 2.275; H5 was supported). Seventh, satisfaction was

significantly associated with brand loyalty (β = 0.349, t = 4.838; H7 was supported). This means that satisfaction with the mixed reality experience has a positive effect on brand loyalty. Table 4 shows all path coefficients and hypotheses test results.

**Table 4.** Path coefficients and hypothesis testing: results of bootstrap analysis.

| | Path name | Original Sample (O) | Sample Mean (M) | Standard Deviation (STDEV) | Standard Error (STERR) | T Statistics (\|O/STERR\|) | Accept/ Reject |
|---|---|---|---|---|---|---|---|
| H1-1 | Interactivity -> Perceived immersion | 0.320 | 0.318 | 0.061 | 0.061 | 5.222*** | **Accept** |
| H1-2 | Interactivity -> Perceived enjoyment | 0.235 | 0.234 | 0.058 | 0.058 | 4.049*** | **Accept** |
| H1-3 | Interactivity ->Brand awareness | 0.240 | 0.237 | 0.070 | 0.070 | 3.447*** | **Accept** |
| H2-1 | Vividness -> Perceived immersion | 0.330 | 0.333 | 0.063 | 0.063 | 5.264*** | **Accept** |
| H2-2 | Vividness -> Perceived enjoyment | 0.395 | 0.399 | 0.062 | 0.062 | 6.346*** | **Accept** |
| H2-3 | Vividness -> Brand awareness | 0.246 | 0.249 | 0.073 | 0.073 | 3.364*** | **Accept** |
| H3-1 | Perceived immersion -> Satisfaction | 0.324 | 0.314 | 0.090 | 0.090 | 3.614*** | **Accept** |
| H3-2 | Perceived immersion -> Brand association | 0.337 | 0.331 | 0.077 | 0.077 | 4.394*** | **Accept** |
| H4-1 | Perceived enjoyment -> Satisfaction | 0.535 | 0.545 | 0.090 | 0.090 | 5.965*** | **Accept** |
| H4-2 | Perceived enjoyment -> Brand association | 0.195 | 0.198 | 0.066 | 0.066 | 2.936** | **Accept** |
| H5 | Brand awareness ->Brand association | 0.162 | 0.163 | 0.071 | 0.071 | 2.275* | **Accept** |
| H6 | Brand association -> Brand loyalty | 0.218 | 0.219 | 0.082 | 0.082 | 2.654** | **Accept** |
| H7 | Satisfaction -> Brand loyalty | 0.349 | 0.348 | 0.072 | 0.072 | 4.838*** | **Accept** |

Note: ***$p < 0.001$ (t > 3.30), **$p < 0.01$ (t > 2.58), *$p < 0.05$ (t > 1.96).

## 6. Discussions

The aim of this study was twofold. The first aim was to investigate how the characteristics of mixed reality (interactivity, vividness) affect the affective aspects (perceived immersion, perceived enjoyment) of the experience and satisfaction of visitors to a mixed reality-based attraction. The second aim was to examine the effects of characteristics of mixed reality on brand awareness, brand association, and, eventually, brand loyalty, which is connected to the management of cultural and artistic visitor attractions. Overall, the findings support our proposed hypotheses. Firstly, the findings show that satisfaction positively influences brand loyalty, which is consistent with previous satisfaction-related research suggesting that online shopping malls have a positive influence on brand loyalty [107] and research on hotel restaurants [108]. The relationship between brand association and brand loyalty was also confirmed, which provides additional, context-specific evidence for the findings of existing studies related to luxury fashion brands [130] and behavior indicating brand loyalty [102]. The same tendencies are evident even in the context of souvenirs and the brand of artistic and cultural attractions that incorporate mixed reality, at least in the case study described herein.

Consistent with prior marketing [99] and luxury car brand studies [100], brand awareness was found to have a positive effect on brand association. However, this study's findings reflect new brand-related factors in a research model including souvenirs as part of the product brand. Perceived immersion had a positive effect on satisfaction, which extends the findings of previous

AR [84] and VR [85] research to the mixed reality context. Previously, perceived immersion was considered as a factor influencing satisfaction or continuance intention [131]; however, this study's unique contribution is to demonstrate the influence of perceived immersion on brand association in the mixed reality context. This influence suggests that perceived immersion is a key factor in sustaining experience and affecting satisfaction for those in charge of developing mixed reality experiences. Additionally, previous research suggests that perceived enjoyment influences satisfaction [90–92], and this relationship was confirmed in this study focusing on mixed reality. Perceived enjoyment was also found to influence the souvenir brand association, which may inform designers of experience venues that management of such exhibitions affects not only the ticket income, but also the souvenir income.

Moreover, studies of interactive games have shown that interactivity has a positive effect on perceived immersion [60] and that interaction between the player and the game world has a positive effect on perceived immersion [62]. With mixed reality technology, interactivity also affects perceived immersion. Therefore, in designing mixed reality experiences, interactivity is an important factor affecting perceived immersion. In this study, interactivity also had a positive effect on perceived enjoyment [62–64]. Hence, when developing technologies or content involving mixed reality, interactivity should be well integrated in order to influence perceived enjoyment positively. Most previous studies on interactivity observe its influence on satisfaction or continuance intention [132,133]. The results of this study also demonstrated that interactivity has a positive effect on brand awareness. For future studies on souvenir purchasing in the field of tourism, it is necessary to design a research model considering brand awareness.

Similar to prior researchers, we found that vividness influenced perceived immersion [42] and perceived enjoyment [74,75]. Perceived immersion was positively influenced by vividness in a study of AR technology in e-commerce [42] and of the learning effect in virtual game worlds [134]. The same relationship was confirmed in our research on the mixed reality technology visitor experience. Furthermore, another study revealed that vividness, interactivity, and telepresence influenced the flow experience of sport spectators engaged in a VR experience [135] and that interactivity and vividness affected consumers' emotional responses and behaviors in video game situations. This study is the first to propose a research model including vividness of experience content in tourist attractions and its effect on brand awareness. Overall, these findings suggest that vividness is an important factor for perceived immersion and perceived enjoyment and should therefore be considered in the development of mixed reality content.

## 7. Conclusions

### 7.1. Theoretical Contributions

There are several theoretical contributions in the current study. First, we used brand equity theory to examine factors influencing satisfaction and brand loyalty of mixed reality experiences at cultural and artistic visitor attractions. Second, looking at characteristics of mixed reality, we suggested that interactivity and vividness not only affect perceived immersion and perceived enjoyment, but also positively affect brand awareness, brand association, and brand loyalty. To the best of the authors' knowledge, this is one of the first studies to employ brand equity theory to examine the effects of characteristics of mixed reality on these constructs in the context of cultural and artistic attractions. We also empirically examined how the technical characteristics of mixed reality influence mixed reality experience (perceived immersion and perceived enjoyment) and visitor satisfaction in the context of a specific cultural and artistic exhibition. We went beyond the boundary of existing research, which focuses on development of experimental environments or proof of concept [13]. Finally, this study adds to existing research (e.g., [5,36,136] on the use of immersive technologies (AR, VR, MR) in cultural heritage organizations, specifically the influence of mixed reality on visitor experience [47].

*7.2. Practical Implications*

This study has several interesting practical implications that can be drawn from the findings. First, the importance of immersion and enjoyment of mixed reality content is emphasized in this study, which is useful for curators and developers of cultural and artistic attractions. Second, our findings associated with factors affecting brand awareness, brand association, and brand equity could help managers of cultural and artistic attractions to develop and create mixed reality content that is closely linked to brand equity. This indicates that enhancing the visitor experience using new and innovative technologies, such as mixed reality, could lead to positive outcomes such as increased visitor numbers, thereby contributing to economic aspects of sustainability, which is particularly important for curators to develop appropriate mixed reality contents in order to increase brand awareness, brand association, and brand loyalty (souvenir purchase). Third, when creating content, practitioners could consider the characteristics of mixed reality highlighted in this study to create immersive visitor experiences. This study demonstrates that perceived immersion and perceived enjoyment influence satisfaction strongly; therefore, practitioners should focus on creating mixed reality content that provides immersive and enjoyable experiences to contribute to the overall sustainability of the cultural and artistic visitor attraction. Although mixed reality content is being developed, it is not often used in practice. L'atelier is one of a few real-world examples where such modern technology has been integrated into a cultural and artistic visitor attraction. With a 17 billion Won investment for its mixed reality exhibition hall development, it is considered a large-scale production cost requiring continuous management performance, which serves as a useful example for similar attractions aiming to produce mixed reality content to enhance the visitor experience. Therefore, rather than only exploring those elements that visitors are satisfied with, we also aimed to examine the influence on brand loyalty and purchase intention, which contribute to increased visitor numbers and revenue generation. This is critical from economic aspects of sustainability as well as for continued operation of cultural heritage attractions.

*7.3. Limitations and Future Research*

This study also has some limitations that provide avenues for further research. First, although the sample was sufficient for the purposes of this study and allowed reasonable conclusions to be drawn, respondents were mostly aged between 20 and 30 years old, which could be considered a limitation. However, as the data were collected with actual visitors to a specific cultural and artistic attraction, this demographic could be considered representative of those visitors who are most likely to use mixed reality technology when visiting this type of attraction. Nevertheless, new studies could include larger and more representative samples within this context to strengthen the current findings. Second, different mixed reality experiences were available for visitors to experience; there was no way of controlling which ones were included in the study. In addition, data were collected at the end of the experience; thus, we relied on respondents recalling their mixed reality experience after the fact. Further quantitative data (e.g., eye-tracking) could be gathered from respondents during the mixed reality experience to provide more accurate responses to specific aspects of the experience and complement the survey results. Additionally, future research could extend this study's quantitative findings by incorporating qualitative measures to obtain in-depth understanding and insight into visitor perspectives of mixed reality experience in cultural and artistic attractions.

**Author Contributions:** Conceptualization, S.B.; methodology, T.H.J. and O.K.; validation, N.M.; formal analysis, S.B. and T.H.J.; data curation, S.B.; writing—original draft preparation, S.B. and T.H.J.; writing—review and editing, N.M. and M.S.; project administration, O.K.; funding acquisition, O.K. All authors have read and agreed to the published version of the manuscript.

**Funding:** This work was supported by a grant from Kyung Hee University in 2019. (KHU-20191209)

**Conflicts of Interest:** The authors declare no conflict of interest.

# Appendix A

**Table A1.** Exploratory factor analysis.

| Variable | Items | Factor Loadings | | | | | | | | Eigen Value | Explained Variance (%) | Confidence Coefficient |
|---|---|---|---|---|---|---|---|---|---|---|---|---|
| | | 1 | 2 | 3 | 4 | 5 | 6 | 7 | 8 | | | |
| **Satisfaction (Sat)** | Sat 1 | **0.804** | 0.217 | 0.121 | 0.137 | 0.316 | 0.155 | 0.215 | 0.213 | 3.486 | 13.942 | 0.978 |
| | Sat 4 | **0.803** | 0.229 | 0.098 | 0.137 | 0.303 | 0.175 | 0.217 | 0.216 | | | |
| | Sat 3 | **0.797** | 0.218 | 0.121 | 0.165 | 0.292 | 0.164 | 0.230 | 0.226 | | | |
| | Sat 2 | **0.791** | 0.234 | 0.141 | 0.149 | 0.270 | 0.153 | 0.227 | 0.255 | | | |
| Brand association (BAs) | BAs 2 | 0.215 | **0.895** | 0.122 | 0.112 | 0.139 | 0.153 | 0.141 | 0.152 | 2.959 | 11.836 | 0.974 |
| | BAs 1 | 0.201 | **0.892** | 0.099 | 0.120 | 0.100 | 0.126 | 0.204 | 0.125 | | | |
| | BAs 3 | 0.208 | **0.873** | 0.137 | 0.172 | 0.182 | 0.141 | 0.165 | 0.146 | | | |
| Interactivity (Int) | Int 3 | 0.056 | 0.148 | **0.915** | 0.087 | 0.067 | 0.140 | 0.091 | 0.092 | 2.904 | 11.617 | 0.958 |
| | Int 2 | 0.133 | 0.072 | **0.912** | 0.142 | 0.110 | 0.154 | 0.142 | 0.141 | | | |
| | Int 1 | 0.123 | 0.097 | **0.900** | 0.088 | 0.146 | 0.121 | 0.161 | 0.145 | | | |
| Brand awareness (BAw) | BAw 2 | 0.131 | 0.119 | 0.136 | **0.897** | 0.110 | 0.222 | 0.141 | 0.096 | 2.832 | 11.328 | 0.946 |
| | BAw 3 | 0.138 | 0.135 | 0.100 | **0.896** | 0.082 | 0.266 | 0.113 | 0.104 | | | |
| | BAw 1 | 0.128 | 0.133 | 0.100 | **0.796** | 0.144 | 0.298 | 0.124 | 0.115 | | | |
| Perceived enjoyment (PE) | PE 2 | 0.329 | 0.175 | 0.140 | 0.123 | **0.816** | 0.156 | 0.201 | 0.141 | 2.733 | 10.933 | 0.959 |
| | PE 4 | 0.323 | 0.168 | 0.151 | 0.143 | **0.816** | 0.128 | 0.197 | 0.195 | | | |
| | PE 3 | 0.342 | 0.133 | 0.118 | 0.128 | **0.803** | 0.127 | 0.222 | 0.199 | | | |
| Brand loyalty (BL) | BL 3 | 0.103 | 0.148 | 0.152 | 0.304 | 0.125 | **0.855** | 0.114 | 0.046 | 2.725 | 10.899 | 0.943 |
| | BL 2 | 0.177 | 0.168 | 0.186 | 0.219 | 0.093 | **0.842** | 0.131 | 0.078 | | | |
| | BL 1 | 0.180 | 0.109 | 0.137 | 0.315 | 0.145 | **0.834** | 0.139 | 0.117 | | | |
| Perceived immersion (PI) | PI 4 | 0.238 | 0.209 | 0.176 | 0.121 | 0.194 | 0.174 | **0.852** | 0.103 | 2.496 | 9.984 | 0.937 |
| | PI 5 | 0.250 | 0.217 | 0.157 | 0.169 | 0.178 | 0.189 | **0.837** | 0.138 | | | |
| | PI 2 | 0.276 | 0.187 | 0.203 | 0.189 | 0.306 | 0.066 | **0.683** | 0.226 | | | |
| Vividness (Viv) | Viv 3 | 0.179 | 0.102 | 0.060 | 0.064 | 0.112 | 0.071 | 0.144 | **0.825** | 2.462 | 9.847 | 0.832 |
| | Viv 2 | 0.138 | 0.165 | 0.172 | 0.052 | 0.129 | 0.159 | 0.084 | **0.811** | | | |
| | Viv 1 | 0.253 | 0.106 | 0.150 | 0.186 | 0.170 | -0.028 | 0.093 | **0.757** | | | |

Note: KMO (Kaiser–Meyer–Olkin measure of sample adequacy) = 0.905, total variance = 90.387%, Bartlett's test of sphericity = 8022.512 (df = 300, $p$ = 0.000).

## Appendix B

**Table A2.** Cross-loading analysis.

| Item | Interactivity | Vividness | Perceived Immersion | Perceived Enjoyment | Brand Awareness | Brand Association | Satisfaction | Brand Loyalty |
|------|---------------|-----------|---------------------|---------------------|-----------------|-------------------|--------------|---------------|
| V01 | **0.964** | 0.368 | 0.453 | 0.397 | 0.305 | 0.327 | 0.384 | 0.372 |
| V02 | **0.978** | 0.368 | 0.438 | 0.381 | 0.353 | 0.311 | 0.385 | 0.410 |
| V03 | **0.941** | 0.298 | 0.366 | 0.302 | 0.285 | 0.326 | 0.296 | 0.360 |
| V04 | 0.325 | **0.872** | 0.398 | 0.449 | 0.326 | 0.344 | 0.516 | 0.232 |
| V05 | 0.352 | **0.870** | 0.381 | 0.405 | 0.281 | 0.374 | 0.465 | 0.315 |
| V06 | 0.259 | **0.855** | 0.377 | 0.390 | 0.252 | 0.322 | 0.462 | 0.237 |
| V07 | 0.428 | 0.477 | **0.902** | 0.624 | 0.425 | 0.486 | 0.638 | 0.382 |
| V08 | 0.409 | 0.371 | **0.960** | 0.554 | 0.390 | 0.495 | 0.593 | 0.435 |
| V09 | 0.401 | 0.406 | **0.963** | 0.555 | 0.436 | 0.511 | 0.611 | 0.462 |
| V10 | 0.364 | 0.432 | 0.588 | **0.960** | 0.380 | 0.459 | 0.702 | 0.422 |
| V11 | 0.347 | 0.475 | 0.594 | **0.958** | 0.379 | 0.428 | 0.710 | 0.393 |
| V12 | 0.379 | 0.478 | 0.590 | **0.967** | 0.394 | 0.459 | 0.707 | 0.407 |
| V13 | 0.305 | 0.320 | 0.424 | 0.395 | **0.911** | 0.372 | 0.421 | 0.594 |
| V14 | 0.331 | 0.313 | 0.432 | 0.382 | **0.966** | 0.363 | 0.414 | 0.565 |
| V15 | 0.300 | 0.314 | 0.409 | 0.362 | **0.971** | 0.374 | 0.413 | 0.595 |
| V16 | 0.300 | 0.367 | 0.518 | 0.419 | 0.357 | **0.968** | 0.529 | 0.381 |
| V17 | 0.325 | 0.398 | 0.496 | 0.451 | 0.362 | **0.978** | 0.551 | 0.409 |
| V18 | 0.348 | 0.405 | 0.530 | 0.492 | 0.417 | **0.978** | 0.567 | 0.422 |
| V19 | 0.358 | 0.528 | 0.628 | 0.722 | 0.414 | 0.539 | **0.971** | 0.449 |
| V20 | 0.381 | 0.563 | 0.638 | 0.699 | 0.426 | 0.557 | **0.966** | 0.455 |
| V21 | 0.364 | 0.540 | 0.640 | 0.712 | 0.441 | 0.544 | **0.971** | 0.464 |
| V22 | 0.342 | 0.528 | 0.626 | 0.715 | 0.418 | 0.548 | **0.970** | 0.461 |
| V23 | 0.369 | 0.322 | 0.449 | 0.433 | 0.617 | 0.384 | 0.480 | **0.955** |
| V24 | 0.394 | 0.284 | 0.432 | 0.387 | 0.540 | 0.410 | 0.454 | **0.939** |
| V25 | 0.364 | 0.246 | 0.403 | 0.381 | 0.594 | 0.385 | 0.404 | **0.949** |

Note: Cross-loadings for each item are higher than those for other constructs, and bold values are above the recommended cutoff value of 0.5 [126].

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
