# Peer review of "The Influence of Mixed Reality on Satisfaction and Brand Loyalty in Cultural Heritage Attractions: A Brand Equity Perspective"

_sustainability, doi:10.3390/su12072956_

Round 1

Reviewer 1 Report

This paper describes a study that tries to identify factors that affect satisfaction with mixed reality experiences at cultural and artistic attractions and their influence on brand loyalty. Authors use brand equity theory to frame their study and investigate the effects of mixed reality on brand awareness, association, and loyalty.

Authors make a good description of background theories and related work, and a good explanation of the hypothesis on which their research model was developed.

I found, however, a few shortcomings in the methods and their description, and on the practical application of the knowledge resulting from this study. The description of the attractions and data collection could be greatly improved as well as the discussion of how the results from this study could be transferred and applied in practice.

The first issue is the description of the cultural and artistic visitor attractions, which could be greatly improved. The mixed reality features employed in the attractions are not clear: why are they used, what did users see, how did users interact, what kind of physical setup do they have? I think that an analysis focused on the interactivity, vividness, and immersion of these mixed reality features would be very useful to understand the visitor experience.

Another issue is how the survey items were constructed or adapted. Authors state that items were adapted from previous literature, but the actual items are never presented. In some of the cited literature, it is hard to understand what was used from that source. Readers are left wondering how the items were worded and how they related to the attractions.

Another issue is the sampling. The description presented in section 4.3 is a bit confusing to me. Authors state that a stratified sampling was applied, but it is not clear how this was done. The population is not described nor the sizes of each strata. It is not even clear to me what strata were considered. It is also unclear how the sampling at the exit of the attractions was performed. From the results presented in table 1, it seems that no random sampling was done. (Sentence on line 327/328 is hard to understand).

I would also appreciate a more in-depth description of how the responses were gathered. Given that a large number of responses were obtained in just two weekends it would be interesting to know how many researchers were conducting the surveys, how long it took to fill each survey, ...

I final issue if the possible practical application of the knowledge generated  by the study. Authors state that "the importance of immersion and enjoyment of mixed reality content . . . is useful for content creators and developers of cultural and artistic attractions". It is not clear to me how content creators can find this results actionable. Given that what makes an attraction more interactive, more immersive, or more enjoyable was not studied. Similarly, how can managers develop mixed reality content that is "closely linked  to brand equity"? Or how can practitioners focus on "creating mixed reality content that provides immersive and enjoyable experiences"?

Author Response

Reviewer 1

Comments

Author Response

The first issue is the description of the cultural and artistic visitor attractions, which could be greatly improved.

Thanks for your comment. We have provided additional description of the cultural and artistic visitor attraction in our revision in response to your comment.

Added:

The attraction, L'atelier, includes artwork and stories of various 19th-century Impressionist artists such as van Gogh and Gauguin. It has recently integrated mixed reality technology to provide an immersive experience using multi-sensory features that contribute to a more interactive and enjoyable visitor experience. By using this technology, visitors can experience not only paintings but also hidden stories about various artists and the historical and cultural background relating to the paintings.

The mixed reality features employed in the attractions are not clear: why are they used, what did users see, how did users interact, what kind of physical setup do they have?

Thanks for your comment. For clarity, we have added mixed reality features employed in the attractions including the reason why they were used, features of mixed reality that visitors will see and interact with, as well as the type of physical setup in the attraction.

Added:

It has recently integrated mixed reality technology to provide an immersive experience using multi-sensory features that contribute to a more interactive and enjoyable visitor experience.

Through AI integration, visitors are able to interact via conversation with friends of the impressionists such as van Gogh’s friend, postman Joseph Roulin. They could also experience artwork through a media art show on the theme of "Monet's Garden” (e.g. visit 360 degrees, and models such as trees in the actual garden appear), which was facilitated by a physical display wall used to portray holograms and projection mapping technology. There is also a musical performance based on van Gogh's life story. In the musicals, actors and characters in digital works interact with real actors through dialogue.

I think that an analysis focused on the interactivity, vividness, and immersion of these mixed reality features would be very useful to understand the visitor experience.

Thanks for your favourable comment.

Another issue is how the survey items were constructed or adapted. Authors state that items were adapted from previous literature, but the actual items are never presented. In some of the cited literature, it is hard to understand what was used from that source. Readers are left wondering how the items were worded and how they related to the attractions.

Thanks for your constructive comment. We acknowledged that it was unclear how the survey items were constructed or adapted as you mentioned. In response to your constructive comment, we have provided an appendix which presents the actual survey items used in this study (see appendix C)

Added:

4.2. Measures

A survey with a total of 25 measurement items was used for this study. The following items were adopted from previous literature: interactivity (e.g. “When I experienced Monet’s garden, the mixed reality water lily leaves interacted with my movement.”) [101], vividness (e.g. “When I experienced a musical, the digital characters felt vivid.”) [102, 103], perceived immersion (e.g. “While I was experiencing the mixed reality content such as a musical and Monet’s garden, I was immersed in the mixed reality experience.”) [104, 105, 72], perceived enjoyment (e.g. “I enjoyed my experience of mixed reality such as the musicals and Monet’s garden.”) [106, 107], brand awareness (e.g. “When I went to L'atelier souvenir shop, I noticed a souvenir that reflects Impressionist artwork.”) [108], brand association (e.g. “When I looked at impressionist souvenirs, I could easily recall my mixed reality experience at L'atelier.”) [109, 110], satisfaction (e.g. “I am satisfied with the mixed reality content I experienced at L’atelier.”) [111, 112], and brand loyalty (e.g. “I am willing to purchase souvenirs from L'atelier souvenir shop.”) [113, 111]. The other items were identified through factor and reliability analyses and scored on a 7-point Likert scale.

Another issue is the sampling. The description presented in section 4.3 is a bit confusing to me. Authors state that a stratified sampling was applied, but it is not clear how this was done. The population is not described nor the sizes of each strata. It is not even clear to me what strata were considered.

Thanks for your keen observation. In response to your comment, we added further information about sampling strategy including population, sizes of each strata, and what strata were considered

Added:

More specifically, in order to identify the population and profile of visitors, researchers consulted with the marketing manager at the visitor and artistic cultural attraction in Korea. According to data gathered between November 2017 and October 2019, the majority of visitors to L'atelier are female (66% female, 34% male) aged between 20 and 30 years (86.7%) followed by those aged over 40 (13.3%). This was also proved during the pilot study. Based on this, sample data were collected by researchers using stratified sampling method to match the population of actual visitors in terms of gender and age.    

It is also unclear how the sampling at the exit of the attractions was performed. From the results presented in table 1, it seems that no random sampling was done. (Sentence on line 327/328 is hard to understand – is this about Table 1).

Thanks for your comment and we have revised the sentence for clarity.

Added:

Table 1 shows the demographic characteristics of the participants which were collected at the exit of the attraction using stratified sampling method.

I would also appreciate a more in-depth description of how the responses were gathered. Given that a large number of responses were obtained in just two weekends it would be interesting to know how many researchers were conducting the surveys, how long it took to fill each survey...

Thanks for valuable comment. We have provided further explanation on how data were collected during two weekends. A total of six researchers were conducting the survey and each participant spent approximately ten minutes completing the survey.

Added:

A total of six researchers were conducting survey data collection during this time.

Each participant spent approximately ten minutes completing the survey.

I final issue if the possible practical application of the knowledge generated by the study. Authors state that "the importance of immersion and enjoyment of mixed reality content . . . is useful for content creators and developers of cultural and artistic attractions". It is not clear to me how content creators can find this results actionable. Given that what makes an attraction more interactive, more immersive, or more enjoyable was not studied.

Thanks for your comment. In response to your comments, we have added additional sentences on the importance and implication of mixed reality contents to content creators.

In this context, content creators are those who plan and design contents in the cultural and artistic attraction. Therefore, it is critical that they understand the importance of immersion and enjoyment of mixed reality content to enhance visitor experience. To avoid confusion, we have changed content creators to curator.

Similarly, how can managers develop mixed reality content that is "closely linked to brand equity"? Or how can practitioners focus on "creating mixed reality content that provides immersive and enjoyable experiences"?

As this is the cultural and artistic visitor attraction context, enhancement of experience using new and innovative technologies such as mixed reality could lead to higher brand awareness, brand association and brand loyalty (souvenir purchase).

Added:

This indicates that enhancing the visitor experience using new and innovative technologies such as mixed reality could lead to such positive outcomes, which is particularly important for curators to develop appropriate mixed reality contents in order to increase brand awareness, brand association and brand loyalty (souvenir purchase).

Reviewer 2 Report

Dear Authors
is a very interesting research area combining mixed reality and brand equity. My main comment is an insufficient description of the research methodology. The research questionnaire should be described and items should be shown on the basis of which statistical calculations and conclusions are made. It is essential for the research to list the items that are part of the individual brand equity dimensions. 
Conclusions should be described separately, at the moment they are together with the discussion of results.

Author Response

Reviewer2

Comments

Author Response

My main comment is an insufficient description of the research methodology.

Thanks for your comment and we have added additional contents in the methodology section and hope you are happy with our revised research methodology section.

Added:

4.1. Study Context

The attraction, L'atelier, includes artwork and stories of various 19th-century Impressionist artists such as van Gogh and Gauguin. It has recently integrated mixed reality technology to provide an immersive experience using multi-sensory features that contribute to a more interactive and enjoyable visitor experience. By using this technology, visitors can not only experience paintings but also hidden stories about various artists and the historical and cultural background relating to the paintings. Through AI integration, visitors are able to interact via conversation with friends of the impressionists such as van Gogh’s friend, postman Joseph Roulin. They could also experience artwork through a media art show on the theme of "Monet's Garden” (e.g. visit 360 degrees, and models such as trees in the actual garden appear), which was facilitated by a physical display wall used to portray holograms and projection mapping technology.

4.2. Measures

The following items were adopted from previous literature: interactivity (e.g. “When I experienced Monet’s garden, the mixed reality water lily leaves interacted with my movement.”) [101], vividness (e.g. “When I experienced a musical, the digital characters felt vivid.”) [102, 103], perceived immersion (e.g. “While I was experiencing the mixed reality content such as a musical and Monet’s garden, I was immersed in the mixed reality experience.”) [104, 105, 72], perceived enjoyment (e.g. “I enjoyed my experience of mixed reality such as the musicals and Monet’s garden.”) [106, 107], brand awareness (e.g. “When I went to L'atelier souvenir shop, I noticed a souvenir that reflects impressionist artwork.”) [108], brand association (e.g. “When I looked at impressionist souvenirs, I could easily recall my mixed reality experience at L'atelier.”) [109, 110], satisfaction (e.g. “I am satisfied with the mixed reality content I experienced at L’atelier.”) [111, 112], and brand loyalty (e.g. “I am willing to purchase souvenirs from L'atelier souvenir shop.”) [113, 111].

4.3. Data Collection

More specifically, in order to identify the population and profile of visitors, researchers consulted with the marketing manager at the visitor and artistic cultural attraction in Korea. According to data gathered between November 2017 and October 2019, the majority of visitors to L'atelier are female (66% female, 33% male) aged between 20 and 30 years (86.7%) followed by those aged over 40 (13.3%). This was also proved during the pilot study. Based on this, sample data were collected by researchers using stratified sampling method to match the population of actual visitors in terms of gender and age.

Each participant spent approximately ten minutes completing the survey. Table 1 shows the demographic characteristics of the participants which were collected at the exit of the attraction using stratified sampling method.

The research questionnaire should be described and items should be shown on the basis of which statistical calculations and conclusions are made. It is essential for the research to list the items that are part of the individual brand equity dimensions.

Thanks for your comment and we have added an appendix which presents the actual survey items used in this study (see appendix C).

Conclusions should be described separately, at the moment they are together with the discussion of results.

Thanks for your comment. We have now following structure of paper which separated conclusion from discussion as per your suggestion.

6 Discussion

7 Conclusion

7.1 Theoretical Contribution

7.2 Practical Implications

7.3 Limitations and Future Research

Appendix C. Survey Items.

Variable

Items

References

Interactivity

(Int)

When I experienced Monet’s garden, the mixed reality water lily leaves moved well to my movement.

[101]

When I experienced Monet’s garden, the mixed reality water lily leaves interacted with my movement.

When I experienced Monet’s garden, the mixed reality water lily leaves moved as I wanted.

Vividness

(Viv)

When I experienced a musical, the digital characters of the musical were clearly visible.

[102, 103]

When I experienced a musical, the digital characters felt vivid.

When I experienced a musical, the digital characters and actor’s voice sounded vivid.

Perceived immersion

(PI)

While I was experiencing the mixed reality experience such as a musical and Monet’s garden, I was absorbed in the mixed reality experience.

[104, 105, 72]

While I was experiencing the mixed reality experience such as a musical and Monet’s garden, I flowed through the mixed reality experience.

While I was experiencing the mixed reality content such as a musical and Monet’s garden, I was immersed in the mixed reality experience.

Perceived enjoyment

(PE)

I enjoyed my experience of mixed reality such as the musicals and Monet’s garden.

[106, 107]

I was interested in my experience of mixed reality such as the musicals and Monet’s garden.

My experience of mixed reality such as the musicals and Monet’s garden were fun.

Brand awareness

(BAw)

When I went to the L'atelier souvenir shop, I noticed a souvenir that reflects impressionist artwork.

[108]

When I went to the L'atelier souvenir shop, it was easy to find souvenirs that reflect impressionist works.

When I went to the L'atelier souvenir shop, the souvenir reflecting the impressionist artwork caught my eye.

Brand association

(BAs)

As I looked at the impressionist souvenirs on display at L'atelier, I remembered the mixed reality experience.

[109, 110]

When I looked at impressionist souvenirs in the souvenir shop, the mixed reality experience at L'atelier was reminiscent.

When I looked at impressionist souvenirs, I could easily recall my mixed reality experience at L'atelier.

Satisfaction

(Sat)

I am satisfied with the mixed reality content I experienced at L’atelier.

[111, 112]

The mixed reality content I experienced at L’atelier were excellent.

Overall, I am satisfied with the mixed reality content I experienced at L’atelier.

The mixed reality content I experienced at L’atelier were good.

Brand loyalty

(BL)

I want to buy a souvenir from the souvenir shop at L'atelier.

[113, 111]

I want to recommend my acquaintances to buy souvenirs from the L'atelier souvenir shop.

I am willing to purchase souvenirs from the L'atelier souvenir shop.

Reviewer 3 Report

"become more and more ubiquitous" I find the concept of "ubiquitous" toooo general and not providing sufficient content. Therefore I recommend that you are more specific.

"sustainability of visitor 36 attractions" What is that? How can attractions be sustainable? Please be more accurate!

"For sustainability and continued operation of cultural and artistic visitor attractions"   It is not clear what you mean. What does sustainability mean in this context? How are visitors attractions sustainable? Please be more accurate!

The abstract is not ok, as the title does not reflect the content of the abstract and vice versa. The title + abstract + key words should be homogeneous and give very exact the content of the paper. Unfortunately it is not the case in this paper. If you present "cultural heritage" in the paper, you should speak of that in the title, abstract + key words. The Research question should be focused on that too!

The introduction should better highlight the research gap and explain how the research gap is transposed into the research question. Than, the authors should explain how their research / paper is original / novel based on the research question. They should further connect the research question to the theory (this is done, maybe a little bit to extensive) but also explain how the research question is implemented in the paper, especially in the research methodology section. How is the empirical research conducted? What are the main implications of the paper?

The introduction should end with a brief presentation of the next sections of the paper.

 Lit review

"Museums and other culture-related curated spaces " which spaces? A bar or a restaurant is also competing for the time of people. Also a mall. Depends on what is relevant for every one of us. So therefore please be more accurate. 

It is interesting that you introduce now the concept of "museum" ... as you have not mentioned this concept before. In the lit review section you present museums, but you have not mentioned that before. Why? You should have spoken from the beginning of this aspects

3. Hypothesis. When presenting the hypothesis and the model... it is recommendable to start with the beginning and not with the end. I mean to start with the first concepts / dimensions on the left ... and than going to the right ... instead you start with the end ... by explaining that there is a relation between satisfaction and loyalty. Ok, I agree that there might be, but how about with the other aspects? 

Study context

Why have you chosen that study context and not another one? Please state some more arguments! You should also insert / include some references for that study context! There are several information, like " L'atelier, includes artwork and stories of various 19th-century Impressionist artists such 297
as van Gogh and Gauguin"  or "Monet's Garden” (e.g. visit 360 degrees, and 304
models such as trees in the actual garden appear), which was facilitated " which are not quite general known. These aspects need to be proper referenced.

"tal of 25 measurement items was used ", yes it is ok to have such measurement items, but it would be better and more accurate to have scales. You explain that you have taken the different items from different references. Ok. Why? Is there no scale that you could have used?

"re was created in English and then translated to 323
Korean by researchers who are proficient in both langu" have you also used translation and back translation in order to ensure the full equivalence of the concepts? Usually when doing such a thing you should have used one research group for the original translation and another researcher group for the back translation. 

"itors to L'atelier are female " why can "L'atelier" considered to be a benchmark? Please give some proper reasons!

Ok, but have you ensured that data is reliable and trustworthy and that it can be interpreted? 

When describing the method you should also describe what analysis you have performed and in which context - with what software etc. 

"alysis. Reliability of the eight identified factors was 368
confirmed by Cronbach's α coefficient, which was higher than 0.832, thus indicating high credibility. 369 5" Ok. References for that?

Table 2 and other tables. Try not to cut them on 2 pages!

Table 3: Correlations. I do not understand why sometimes you use "0.xx" and other times ".xx". It would be best to always write aspects in the same manner!

Table 4: now this is interesting. All hypothesis confirmed? This is just perfect. I have really doubts that all hypothesis can be confirmed! 

6. Discussions with an s

Conclusions with an s.

You should start with 7.1. Theoretical implications. Where is 7.1.???

The paper is ok, but where is the connection with sustainability?

Author Response

Revision Summary

Manuscript ID: sustainability-719771.R2

Title: The Influence of Mixed Reality on Satisfaction and Brand Loyalty: A Brand Equity Perspective

Reviewer 3

Comments

Author Response

"become more and more ubiquitous" I find the concept of "ubiquitous" toooo general and not providing sufficient content. Therefore I recommend that you are more specific.

Thanks for your comment. We have provided additional content for clarity.

Added:

We made the sentence more specific as follows:

As 5G provides very high data rates, extremely low latency, and significant improvement in users' perceived quality of service (QoS), it is expected to become more widespread in the coming years particularly in urban areas (Minoli & Occhiogrosso, 2019), hence, interest in developing mixed reality content with high user acceptance is increasing.

Minoli, D., & Occhiogrosso, B. (2019). Practical Aspects for the Integration of 5G Networks and IoT Applications in Smart Cities Environments. Wireless Communications and Mobile Computing, Article ID 5710834.

"sustainability of visitor attractions" What is that? How can attractions be sustainable? Please be more accurate!

"For sustainability and continued operation of cultural and artistic visitor attractions"   It is not clear what you mean. What does sustainability mean in this context? How are visitors attractions sustainable? Please be more accurate!

Thanks for your comment. We have revised the sentence for clarity and expanded more on the meaning of sustainability. 

Added:

As with any other organizations, the sustainability of cultural and arts visitor attractions consist of four elements including the cultural environment (e.g. heritage preservation, new audiences), social environment (e.g. engagement, social responsibility), natural environment (e.g. green technologies, energy efficiency) and economic environment (Ren & Han, 2018). Economic aspects can be supported by increasing visitor numbers and net growth (Ren & Han, 2018), hence, when considering factors that facilitate the continued operation of the cultural and arts visitor attraction, it is necessary to study factors affecting visitor satisfaction [8] and souvenir purchase [9].

Ren, W., & Han, F. (2018). Indicators for assessing the sustainability of built heritage attractions: An Anglo-Chinese Study. Sustainability, 10, 2504, 1-28.

Stylianou-Lambert, T. Boukas, N. & Christodoulou-Yerali, M. (2014). Museums and cultural sustainability: Stakeholders, forces, and cultural policies. International Journal of Cultural Policy, 20, 566–587.

The abstract is not ok, as the title does not reflect the content of the abstract and vice versa. The title + abstract + key words should be homogeneous and give very exact the content of the paper. Unfortunately it is not the case in this paper. If you present "cultural heritage" in the paper, you should speak of that in the title, abstract + key words. The Research question should be focused on that too!

Thanks for your favourable comment. We added “cultural heritage attractions” in the keywords list and title for homogeneity:

The Influence of Mixed Reality on Satisfaction and Brand Loyalty in Cultural Heritage Attractions: A Brand Equity Perspective.

The introduction should better highlight the research gap and explain how the research gap is transposed into the research question. Than, the authors should explain how their research / paper is original / novel based on the research question. They should further connect the research question to the theory (this is done, maybe a little bit to extensive) but also explain how the research question is implemented in the paper, especially in the research methodology section. How is the empirical research conducted? What are the main implications of the paper?

The introduction should end with a brief presentation of the next sections of the paper.

Thanks for your valuable comment. We have revised the introduction to highlight the research gap and how it is tranposed into the research question. We also highlighted the unique contribution of this study to the brand equity theory in the context of cultural heritage attractions. We also provided how the research question is implemented in the methodology section as well as theoretical implication of this study. We also provided the short sentence at the end of introduction to connect the next section of the paper.

Added:

However, to date, there is limited research employing a brand equity perspective to investigate the influence of visitors’ mixed reality experience on brand equity in the context of cultural and artistic visitor attractions through empirical research using quantitative approach. Therefore, this study aims to bridge this research gap and as a result, it offers important implications for managers of such organizations in addition to making a theoretical contribution to research in this area.

The paper begins with the theoretical background exploring previous literature on the implementation of mixed reality in cultural heritage attractions followed by an overview of brand equity theory. A justification for each of the proposed hypotheses is then presented, which is followed by the method detailing the quantitative data collection and analysis. Finally, discussions and conclusions including theoretical contributions and practical implications are offered.  

Lit review

"Museums and other culture-related curated spaces " which spaces? A bar or a restaurant is also competing for the time of people. Also a mall. Depends on what is relevant for every one of us. So therefore please be more accurate. 

It is interesting that you introduce now the concept of "museum" ... as you have not mentioned this concept before. In the lit review section you present museums, but you have not mentioned that before. Why? You should have spoken from the beginning of this aspects

Thanks for valuable comment. We have revised the below sentence for clarity. We also changed ‘people’ to ‘cultural visitor’

Cultural heritage-related curated spaces including art galleries and museums compete for cultural visitors’ time.

3. Hypothesis. When presenting the hypothesis and the model... it is recommendable to start with the beginning and not with the end. I mean to start with the first concepts / dimensions on the left ... and than going to the right ... instead you start with the end ... by explaining that there is a relation between satisfaction and loyalty. Ok, I agree that there might be, but how about with the other aspects? 

Thanks for your suggestion. In response to your recommendation, we changed the order of the hypotheses. Please see revised section 3.1- 3.7.

Study context

Why have you chosen that study context and not another one? Please state some more arguments! You should also insert / include some references for that study context! There are several information, like " L'atelier, includes artwork and stories of various 19th-century Impressionist artists such 297
as van Gogh and Gauguin"  or "Monet's Garden” (e.g. visit 360 degrees, and 304
models such as trees in the actual garden appear), which was facilitated" which are not quite general known. These aspects need to be proper referenced.

Thanks for your comment. We have added the justification why this particular attraction has been selected. In addition, we have provided relevant references.

Added:

The attraction, L’atelier, is a representative IT-based theme park which substantially and most successfully adopts interactive media including mixed reality technologies. Hence, L’atelier can be a good benchmark to investigate the viability of mixed reality technology for the sustainable operation of the cultural heritage attraction.

Monet depicted the water lily that changes with light, hence, the Artist’s garden in the media art show is the painting created by light (L’atelier, 2020). There is also a Hologram Talk Show based on the mystery around the death of van Gogh and a musical performance - “van Gogh’s Dream” - which is based on his life story (L’atelier, 2020).

L’atelier (2020)

http://www.light-atelier.com/eng/sub/0102.php

Korean team to add reference in-text and reference list.

"total of 25 measurement items was used ", yes it is ok to have such measurement items, but it would be better and more accurate to have scales. You explain that you have taken the different items from different references. Ok. Why? Is there no scale that you could have used?

Thank you for your comment. We acknowledge that there has been no study which contains all items that should appear in our research model. Therefore, we combined items from a number of references to inform our study.

"questionnaire was created in English and then translated to 323
Korean by researchers who are proficient in both langu" have you also used translation and back translation in order to ensure the full equivalence of the concepts? Usually when doing such a thing you should have used one research group for the original translation and another researcher group for the back translation. 

Yes, we have two research groups in Korea and UK. The Korean group interpreted the English items, amended to fit to the context of the study, and then translated them into Korean. Then UK team re-translated the Korean items into English. Then we verified that there is no problem in the translation.

why can "L'atelier" considered to be a benchmark? Please give some proper reasons!

Thanks for your comment. We have provided justification why this particular cultural heritage attraction was selected.

Added:

The attraction, L’atelier, is a representative IT-based theme park which substantially and most successfully adopts interactive media including mixed reality technologies. Hence, L’atelier can be a good benchmark to investigate the viability of mixed reality technology for the sustainable operation of the cultural heritage attraction.

Ok, but have you ensured that data is reliable and trustworthy and that it can be interpreted? 

Thanks for your comment. The researchers followed procedures to ensure reliability and validity of the data. For example, a pilot study and reliability analyses were conducted. 

When describing the method you should also describe what analysis you have performed and in which context - with what software etc. 

Thanks for your comment and we added the following sentence:

Added

SPSS 23.0 was used for demographic analysis, and exploratory factor and reliability analyses.

"alysis. Reliability of the eight identified factors was 368
confirmed by Cronbach's α coefficient, which was higher than 0.832, thus indicating high credibility. 369 5" Ok. References for that?

Thanks for your keen observation. We have newly added a reference (#132) to support our statement in the revision:

Added

132. Bland, J. M., & Altman, D. G. (1997). Statistics notes: Cronbach's alpha. BMJ, 314(7080), 572.

Table 2 and other tables. Try not to cut them on 2 pages!

Thanks for your comment. We acknowledge that it is best to present each table on one page and will ensure the correct format during the final stages of editing which will also be confirmed by the editors.  

Table 3: Correlations. I do not understand why sometimes you use "0.xx" and other times ".xx". It would be best to always write aspects in the same manner!

Thanks for your comment and we used “.xx” in Table 3 for consistency.

Table 4: now this is interesting. All hypothesis confirmed? This is just perfect. I have really doubts that all hypothesis can be confirmed! 

Thanks for your comment. Previous research confirmed that in some cases, all hypotheses in the research model are supported whilst in other cases, not all hypotheses are supported. In this research, authors found that all hypotheses in the research model were supported based on the empirical test.

6. Discussions with an s

Conclusions with an s.

Thanks for your comment and we corrected as per your suggestion:

Discussions and Conclusions.

You should start with 7.1. Theoretical implications. Where is 7.1.???

Thanks for your keen observation and we have corrected numberings.

7.1 Theoretical Contributions’.

The paper is ok, but where is the connection with sustainability?

Thanks for your comment and we have provided additional sentences how outcomes of this study is connected with sustainability.

Added:

Although mixed reality content is being developed, it is not often used in practice. L’atelier is one of few real-world examples where such modern technology has been integrated into a cultural and arts visitor attraction. With a 17 billion Won investment for its mixed reality exhibition hall development, it is considered a large-scale production cost requiring continuous management performance, which serves as a useful example for similar attractions aiming to produce mixed reality content to enhance the visitor experience. Therefore, rather than only exploring those elements that visitors are satisfied with, we also aimed to examine the influence on brand loyalty and purchase intention which contribute to increased visitor numbers and revenue generation. This is critical from economic aspects of sustainability as well as for continued operation of cultural heritage attractions.  

Round 2

Reviewer 2 Report

Dear Authors
The article has been significantly improved, in particular as regards the discussion of results and conclusions. As such, it may be published in Sustainability.

Author Response

The article has been significantly improved, in particular as regards the discussion of results and conclusions. As such, it may be published in Sustainability.

Thanks for your comment which contributed to the improvement of quality of paper.
